# See both ways: A bidirectional evaluation of Multimodal Language Models and Human Spontaneous Speech for Image Captioning

## Abstract

Multimodal large language models (MLLMs) have achieved notable success in image captioning, yet systematic comparisons with human-generated references remain underexplored. In this work, we present a novel study on understanding the alignment between captions generated by multimodal models and spontaneous human speech captions. To this end, we introduce a human–machine bidirectional evaluation framework, extending a recently proposed image-caption evaluation metric. This evaluation is performed by comparing crowd-sourced audio-based captions of images with model generated captions from various MLLMs. Our detailed analysis reveals that, (i) humans are more selective than models in describing specific aspects of the image rather than providing a comprehensive summary, (ii) scores with human reference and model targets are significantly higher than those computed with model reference and human response, and (iii) images from specific categories like "nature" and "educational" evoke more human imagination during the description task, compared to other categories. Together, these findings reveal a clear divergence in human vs. model captioning.

## 1 Introduction

Image captioning, the task of generating a verbal depiction of a visual presentation, is one of the most fundamental human skills. The spontaneous vocal description of images evokes visual attention, linguistic proficiency and memory, and is used in neuro-psychological assessments of cognitive skills (For example, the "Cookie Theft" picture description from the Boston-Diagnostic-Aphasia-Examination (Giles et al., 1996)). In natural image captioning, early work by Griffin & Bock (2000) suggested that eye movements during picture description predict the order of mention. Huettig et al. (2011) explored the use of verbal descriptions of visual objects as a tool to probe planning processes. In non-English settings, a recent study by Takmaz et al. (2024) on Dutch image description, attempted to relate the properties of an image and the human behavior while describing the image to quantify the visuo-linguistic complexity. Separately, He et al. (2019) found that human attention, measured via eye-fixations, differs from regular viewing of images.

In machine learning, the image captioning task is also considered one of the challenging tasks that require visual and contextual understanding, saliency and relationship cognizance, and multi-modal alignment. Early approaches were template-based and more rigid in the caption generation (Farhadi et al., 2010). This was advanced by representation based approaches, like those investigated by Kulkarni et al. (2011) and Yang et al. (2011). As deep learning and sequence modeling architectures became popular, neural encoder-decoder models (for example, Vinyals et al. (2015) and Xu et al. (2015)) showed substantial improvements in free-form caption generation for natural images. Transformer models, like the proposal by Anderson et al. (2018), further advanced this progress through attention based modeling. More recently, multi-modal models (Li et al. (2023a)) and large language models (OpenAI (2024)) have become de facto image-captioning systems, owing largely to their improvements in understanding, reasoning and generation capabilities.

The evaluation of image captioning systems has also witnessed significant improvements. The early frameworks, based on text evaluation metrics like BLEU (Papineni et al., 2002) and approximating human consensus (Vedantam et al., 2015), have paved the way for semantic similarity-based (Anderson et al., 2016) and embedding based (Hessel et al., 2021b) metrics. In the recent efforts, large language models have enabled the evaluation of image captioning systems in a more human-like setting (Ye et al. (2025)).

In this paper, we undertake a study on comparing human and model generated captions on natural, culturally ingrained, and regionally relevant images. The dataset of images and the human captions are derived from Vaani dataset (VAANI (2025)), where human participants provide audio captions of images in Hindi language. The proposed large-scale analysis in this paper, (10k image-transcript pairs and with 3288 human participants), involves the premise where both humans and models provide captions without a "ground-truth" setting. Hence, the study is singularly unique in various ways compared to previous studies.

The key contributions of this paper are:

1. We present, to the best of our knowledge, the first large-scale, comprehensive analysis of the Vaani dataset VAANI (2025), a multilingual, multimodal benchmark consisting of spontaneous, image-prompted speech.

2. We introduce a bidirectional evaluation framework based on two complementary metrics: the Human-as-Reference (HAR) score, which measures how closely model captions align with human references, and the Model-as-Reference (MAR) score, which evaluates the reverse alignment.

3. Using these scores, we conduct a detailed model-wise analysis of both open-source systems (Gemma-3-12B and Llama-4-scout-17b-16e-instruct) and closed-source systems (Gemini 2.5 Pro and GPT-4o). Our study provides novel insights into differences in coverage, selectivity and hallucination profiles across models.

4. We perform a category-wise evaluation of human transcripts, examining how the image-caption quality varies across broad semantic categories. This analysis highlights the differences in how humans and models prioritize visual content.

## 2 RELATED WORK

**Image Captioning Evaluation:** Traditional approaches evaluating image captioning primarily rely on reference-based methods. BLEU (Papineni et al. (2002), CIDEr (Vedantam et al. (2015)), and METEOR (Denkowski & Lavie (2014) emphasize lexical overlap, thereby limiting their ability to capture deeper semantic attributes. Subsequent efforts, such as SPICE (Anderson et al. (2016)), CLIP (Radford et al. (2021)) and BLIP-2 (Li et al. (2023b)) have improved the quality of the metrics. Examples include CLIPScore (Hessel et al., 2021a), PACScore (Sarto et al., 2023), and BLIP2Score (Zeng et al., 2024).

Recent integration of MLLMs into the captioning evaluation pipeline has helped progress from reference-dependent metrics that ignore image content, observed in CLAIR (Chan et al., 2023), to reference-free, image-grounded approaches providing explainable scores (FLEUR (Lee et al., 2024)). However, these explanations often lack a standardized structure and their quality remains unverified. To address this, EXPERT (Kim et al., 2025) introduces a framework that generates structured explanations evaluated on the basis of fluency, relevance, and descriptiveness. A recent effort on image caption evaluation, termed DCScore (Ye et al., 2025), attempted the use of concept level quality evaluation using LLMs. However, a critical limitation of existing vision-language research is the linguistic and cultural bias embedded in canonical datasets. As noted by Liu et al. (2021), this issue stems from an over-reliance on Anglocentric data sources.

**Multimodal Large Language Models:** Vision-Language Models (VLMs) (Dai et al., 2023; Li et al., 2023b; Jia et al., 2021; Yu et al., 2022) have substantially advanced image captioning tasks by developing large-scale, pre-trained highly capable Multimodal Large Language Models (MLLMs) (Bai et al., 2023b; Gao et al., 2023; Achiam et al., 2023; Chen et al., 2024a; Comanici et al., 2025). Models such as LLaVa (Liu et al., 2023b;a; 2024), Qwen-

VL (Bai et al., 2023a), Intern-vl (Chen et al., 2024b) have accelerated the development of these general-purpose models.

Nevertheless, despite these advanced capabilities, MLLMs are susceptible to significant failure modes, including incomplete descriptions and object hallucinations. A major reason for this phenomenon is a fundamental imbalance in their training (Pi et al., 2024; Sun et al., 2024; Yu et al., 2023; 2024). In this paper, we evaluate a suite of contemporary MLLMs on the Vaani dataset to assess the alignment between model- and human-generated captions.

## 3 METHODOLOGY

### 3.1 THE VAANI DATASET

Auditing the cultural awareness of current MLLMs necessitates a dataset that prominently features non-Western contexts. To this end, India's vast linguistic landscape, with hundreds of languages and dialects underrepresented in digital resources, provides a critical testbed. The Vaani dataset (VAANI, 2025), a large-scale, open-source, multilingual, and multimodal corpus, contains approximately $27,751$ hours of spontaneous, image-prompted speech collected from 143K speakers across 145 Indian districts. This includes descriptions of 258K images in 103 languages. Of this audio data, $1,187$ hours have been manually annotated. This work focuses on the Hindi subset, which contains 594 speech hours and constitutes the largest monolingual corpus within the transcribed data. We specifically use its test set, comprising $9,888$ unique image-transcript pairs from 3288 speakers.

For each image, participants were prompted to describe the image in their native language or dialect. The resulting audio recordings are accompanied by the language of the recording, the speaker's full linguistic range (e.g., Hindi, Telugu, English), demographic information (gender, state, and district), and a verbatim transcription of the speech. Unlike traditional captioning datasets that rely on carefully curated, text-based annotations, Vaani captures the way humans describe visual content in real time, often casually, with hesitations or incomplete phrasing. This makes it particularly compelling for evaluating image captioning models in realistic settings, where system outputs can be compared with naturalistic speech-based captions. An example is shown in Table 1.

Table 1: Reference images and Human-generated Captions

| Reference image | Metadata | | Transcript |
|---|---|---|---|
|  Specific_01 | • Language: Hindi
• Languages Known: {Hindi}
• Gender: Female
• Transcription Available: Yes | • State: Bihar
• District: Begusarai
• Pincode: 848210
• Stay (years): NA(21) | <noise> इस फोटो {photo} में सामने की तरफ एक लड़का दिखाई दे रहा है जो स्कूटर {scooter} पे पर बैठा हुआ दिखाई दे रहा है जि–सने ब्लू {blue} कलर {colour} की टी–शर्ट {t-shirt} </noise> |
|  Generic_03 | • Language: Hindi
• Languages Known: {'Hindi', 'Telugu'}
• Gender: Male
• Transcription Available: Yes | • State: Andhra Pradesh
• District: Anantapur
• Pincode: 515581
• Stay (years): Anantapur(22) | <noise> यहां पाये मशीन {machine} लगी हुई हैं अंदर और आदमी कम कर रहे हैं और मशीनों का रंग सफेद हैं और आदमी कला हेला हैं।</noise> |

### 3.2 DCSCORE METRIC

Ye et al. (2025) proposed the DCScore, a novel metric designed to evaluate hallucinations and factual correctness. This involves the following steps:

1. Decomposition: Generated captions and ground-truth captions are decomposed into the smallest standalone facts or self-sufficient units, referred to as Primitive Information Units (PIUs) (Ye et al., 2025), using MLLMs. Human transcripts are broken down into a set of PIUs, denoted as $T = \{t_1, t_2, \ldots, t_M\}$, where $M$ is the number of extracted units. Similarly, model-generated captions are decomposed into a set $P = \{p_1, p_2, \ldots, p_N\}$, where $N$ represents the number of extracted units.

2. Matching: An LLM is prompted to determine whether each primitive unit in $p_i \in P$, from the generated captions, is either explicitly stated or logically inferable from a corresponding unit $t_j \in T$ in the human transcript. The alignment between generated and human captions is defined as $Q = P \cap T$, where $Q$ represents the set of overlapping PIUs.

3. Verification: The PIU in the generated caption is verified against the input image using an LLM. The verification process evaluates the accuracy of each unit $p_i$ in the generated captions $P$ by directly referencing the corresponding image. Following the original DCScore methodology, we employ the version-specific GPT-4.1 (OpenAI, 2025), with stable API access, to guarantee that our evaluation metric is deterministic and fully reproducible across studies.

**Evaluation Metrics**

From the model-generated set $P$, the human transcript set $T$, and their overlap $Q$, precision, recall and F1 scores are defined to evaluate caption quality.

**Precision score** $s_p$ measures the proportion of correct PIUs among those produced by the MLLM:

$$s_p = \frac{|P_{true}|}{|P|} \tag{1}$$

where $P_{true} = \{p_i \mid p_i \in P, \ p_i \text{ is correct}\}$ denotes the subset of correct units in $P$.

**Recall score** $s_r$ measures the proportion of human transcript PIUs that are either directly aligned or correctly produced by the MLLM:

$$s_r = \frac{|Q| + |P_{true} \setminus Q|}{|T| + |P_{true} \setminus Q|}, \tag{2}$$

The overall quality of captioning is measured using the **F1 score** $s_f$, which is the harmonic mean of precision ($s_p$) and recall ($s_r$).

### 3.3 Proposed Bidirectional Evaluation Framework

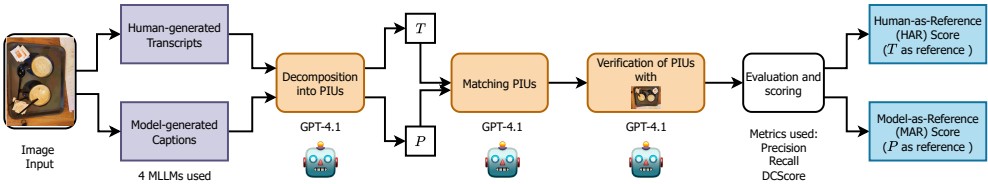

Figure 1: Bidirectional framework to assess the alignment between model- and human-generated captions

Inspired by the DCScore evaluation metric (Ye et al., 2025), we propose a bidirectional framework containing the following parts.

1. **Human-as-Reference (HAR) Score:** Here, the model captions form the target text, which are evaluated against the human transcript reference.

2. **Model-as-Reference (MAR) Score:** Here, the human-transcripts form the target captions, which are evaluated against the model reference.

Figures 2 and 7 (Appendix A.1) provide a concrete example of our bidirectional scoring framework for a single image. For clarity, the results in this figure are translated into English; the original Hindi text is shown as a red underlay.

In the **Human-as-Reference (HAR) evaluation** (Figure 2), the subjective PIUs from the model, such as "पूरी छवि में कैफे जैसा माहौल महसूस होता है" (*The entire image gives the*

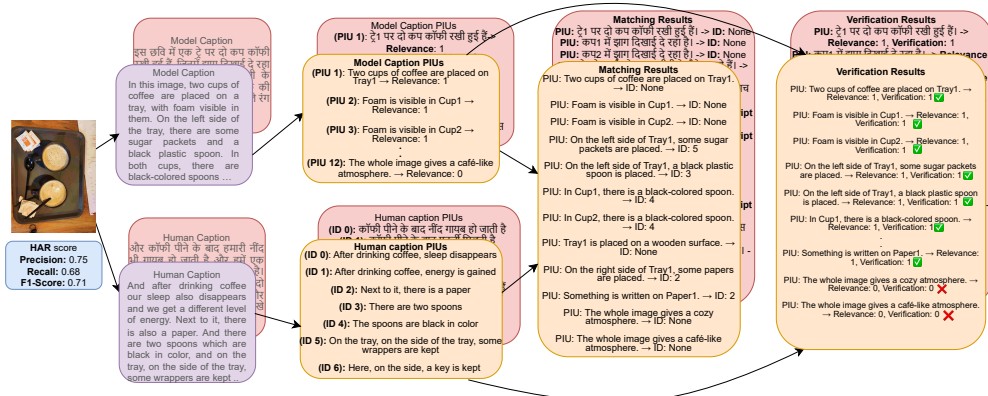

Figure 2: Illustration of Human-as-Reference (HAR) evaluation for a single data point. The corresponding MAR evaluation is depicted in Figure 7.

*feeling of a café-like atmosphere.*), are assigned a relevance score of 0, as they are not direct visual facts. The framework then quantifies the alignment to produce the precision (0.75), recall (0.68), and F1 (0.71). Conversely, the **Model-as-Reference (MAR) evaluation** (Figure 7) highlights how human descriptions can include inferential statements that are not visually grounded in the image. The motivation for the bidirectional evaluation is provided in Sec. A.2.

## 3.4 EXPERIMENTAL SETUP

**Models Evaluated:** Our experimental setup evaluates four MLLMs, selected across sizes and open/closed model families (Table 2). Each model was tasked with generating a caption for every image in our dataset using the standardized prompt: "Describe the image in comprehensive detail as a single paragraph in Hindi."

Table 2: Overview of the MLLMs used in our bidirectional evaluation framework.

| Stage | MLLM | Family | Size | Release |
|---|---|---|---|---|
| Caption Generation | Gemini 2.5 Pro[1] [No Thinking] | Gemini | – | 2025 |
| | GPT-4o[2] [No thinking] | GPT | – | 2024 |
| | Gemma-3-12B[3] | Gemma | 12B | 2024 |
| | Llama-4-scout-17b-16e-instruct[4] | Llama | 17B | 2025 |
| Decomposition Matching & Verification | GPT-4.1[5] [No thinking] | GPT | - | 2025 |

## 4 RESULTS

We evaluate results along four aspects:

1. **Bidirectional scoring**: We examine both Human-as-Reference (HAR) and Model-as-Reference (MAR) evaluation scores to capture asymmetries of evaluation.

2. **Statistical validation:** To assess the robustness of observed differences, we performed an unpaired Welch (1947) t-test, comparing each model with a baseline model (highest F1 score) to determine statistical significance in performance gaps.

---

[1] https://deepmind.google/models/gemini/pro/

[2] https://openai.com/index/hello-gpt-4o/

[3] https://huggingface.co/google/gemma-3-12b-it

[4] https://huggingface.co/meta-llama/Llama-4-Scout-17B-16E

[5] https://openai.com/index/gpt-4-1/

3. **Error Quantification:** We define:

   (a) Hallucination Rate as $1 - $ Precision, i.e., the proportion of incorrect PIUs in a model/human-generated caption.

   (b) The Omission Rate: proportion of reference PIUs, the models/humans fail to capture.

   $$\text{Omission Rate} = \frac{N_{\text{reference}} - N_{\text{matched}}}{N_{\text{reference}}}$$

4. **Sample difficulty:** To further examine the differences at category-level, we analyze sample difficulty by bucketing each image into Easy, Medium, or Hard based on a global HAR/MAR performance threshold.

## 4.1 Model-wise Analysis

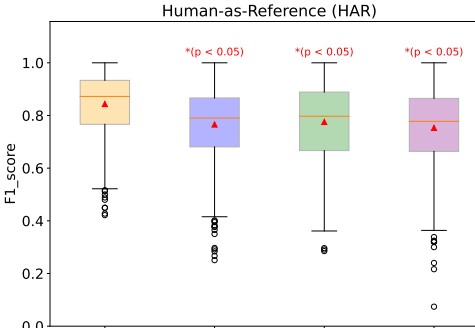 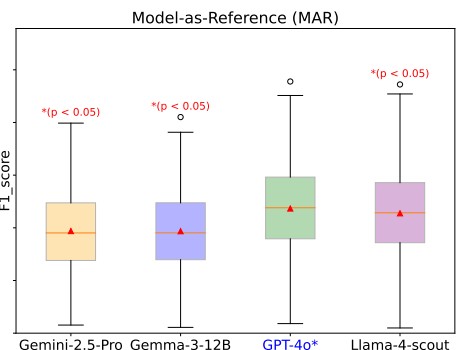

Figure 3: Boxplot visualization of the DCScore (HAR and MAR) of 200 random image samples, across all models. The reference (highest F1 score) model is highlighted in blue on x-axis and models with $p < 0.05$ are significantly different from the reference.

Figure 3 presents model-wise performance under the bidirectional scoring framework for a random set of 200 images. Results indicate that Gemini-2.5-Pro achieves the highest mean F1 score in the HAR setting but comparatively lower F1 in MAR. A high MAR score reflects captions that are more exhaustive, with stronger coverage (Recall), as further illustrated in Figure 13 (Appendix A.5). In contrast, GPT-4o shows the opposite trend, with a comparatively lower F1 score in HAR, but a high F1 score in MAR, suggesting that its captions are less exhaustive yet more closely aligned with human-generated captions compared to the other models. To contextualize these results, precision-recall trade-offs are further analyzed in Figure 13 (Appendix A.5).

For statistical validation, each model was compared with Gemini-2.5-Pro in the HAR evaluation pipeline and GPT-4o in the MAR evaluation pipeline, using unpaired t-tests. In the HAR direction, the differences in F1-score between Gemini-2.5-Pro and the other models (Gemma-3-12B, GPT-4o, and Llama-4-scout) were statistically significant (p < 0.05), demonstrating that Gemini-2.5-Pro is the best performing model statistically. In the MAR direction, GPT-4o significantly outperformed the alternative models.

**Hallucination and Omission Rates:** Beyond F1 differences, hallucination and omission patterns provide deeper insights into model error profiles (Figure 4). In the HAR direction, hallucination rates remain consistently low (0.02–0.16) across models. Gemini-2.5-Pro and GPT-4o maintained the strongest control ($\approx$ 0.02–0.03), while Gemma-3-12B displayed an elevated hallucination rate of 0.12, suggesting a tendency to over-generate details in certain images. Despite low hallucination, omission rates were substantial (0.43–0.51) with GPT-4o having relatively higher omission rate ($\approx$ 0.51), while Gemini (0.43) and Gemma (0.48) performed slightly better. In the MAR direction, omission rates are higher (0.60–0.71), reflecting the challenge for the human-generated captions in terms of coverage when compared to the verbose model captions. Gemma was found to be the most omission-prone ($\approx$ 0.71), while GPT-4o displayed the best balance between omission and hallucination.

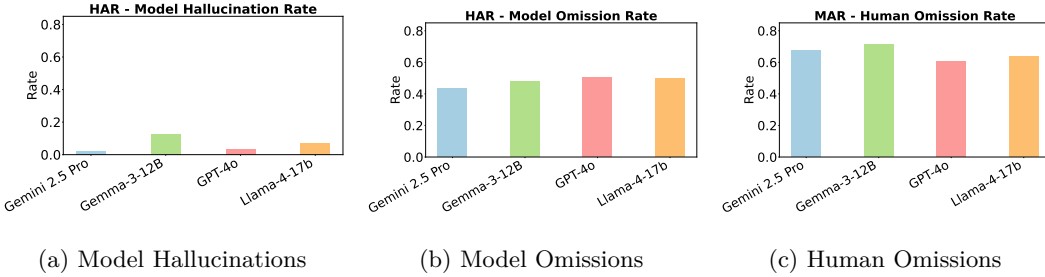

(a) Model Hallucinations     (b) Model Omissions     (c) Human Omissions

Figure 4: Bidirectional hallucination and omission rates: (a) Model hallucinations in the HAR setting, (b) Model omissions in the HAR setting, and (c) Human omissions when model captions serve as reference (MAR)

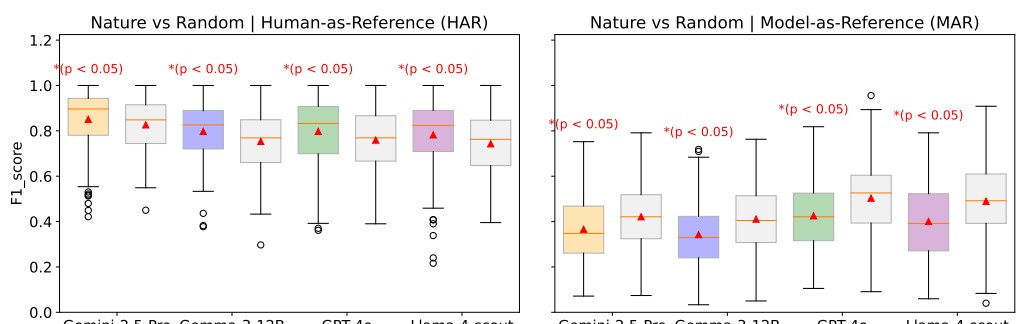

Figure 5: Boxplot visualization of the DCScore (HAR and MAR) across all models for the Nature and Random categories. For Nature, all Models are significantly different ($p < 0.05$) from the Random category. Grey colour represents the Random category across all models, while the Nature category is shown in a distinct colour for each model.

In the MAR setting, where human-generated captions serve as the reference, higher omission rates imply that human annotations often fail to account for all relevant image regions. Figure **??**(c) shows an average omission rate of approximately 70%, suggesting that humans focus only on 30% of the PIUs in an image. In contrast, models exhibit a lower omission rate of around 40%, indicating a reduced tendency to skip visual contents.

From these observations, we conclude that there is a consistent divergence between human- and model-generated captions. Humans demonstrate strong visual selectivity, i.e., they describe only a subset of the image, focusing on salient regions while omitting secondary details. Models, by comparison, generate captions that are denser and more exhaustive, often attempting to cover the entire image. This discrepancy reflects the divergent objectives - humans prioritize communicative sufficiency, whereas models are optimized for coverage.

### 4.2 CATEGORY-WISE ANALYSIS

**Category Selection:** The unique set of $9,888$ images from the Hindi test set was further used for categorization. The prompt and strategy used for this classification is provided in Appendix A.3. Examples are provided in Section A.4. The categories are:

1. **Commercial & Retail:** Scenes of markets, vendors, and trade.

2. **Religion & Culture:** Religious ceremonies, festivals, or cultural heritage.

3. **Nature & Landscape:** Natural elements and wildlife.

4. **Education & Learning:** Academic or instructional settings.

5. **Infrastructure & Transport:** Buildings, public works and modes of transit.

Figure 5 reports the statistical significance results for the Nature category relative to the random reference. In the HAR setting, across all four models, the F1 scores for Nature are significantly higher than random images (p < 0.05), indicating that models consistently describe natural scenes more effectively than random ones. In the MAR setting, F1 scores are overall lower than in HAR but still significantly higher than the random baseline (p < 0.05). This asymmetry reflects the fact that humans are more selective and tend to omit details in certain categories. Figure 20 in Appendix A.5 shows the F1 score across five categories for all four models.

**Hallucination and Omission Rates:** Following the statistical validation from our t-tests and referring to the HAR/MAR Hallucination and Omission rates discussed in Section 4.1, HAR omission rates generally follow visual complexity. Commercial and Infrastructure categories have the highest omissions, averaging around 0.52 and 0.49, reflecting the difficulty of capturing dense scenes. Culture also shows elevated omissions (≈ 0.48), while Random and Educational categories are slightly lower (≈ 0.48 and 0.44). MAR omission rates are higher overall, typically ranging from 0.60 to 0.75. Education and Nature scenes show the highest omissions (≈ 0.72). This reversal in MAR omissions indicates that models tend to be more verbose in Nature and Educational scenes compared to humans.

### 4.2.1 THEMATIC CATEGORY DIFFICULTY DISTRIBUTION

To further investigate these category-level differences, we analyzed the difficulty of individual samples as can be observed in Figure 14 (Appendix A.5). By bucketing each image into 'Easy', 'Medium', or 'Hard' based on an HAR/MAR global F1 threshold, we observed a clear pattern. Specifically, the Nature and Educational categories contain the highest proportion of 'Easy' samples and the lowest proportion of 'Hard' ones.

As seen in an example of an image from the Nature Category, Figure 19 (Appendix A.5), the model caption describes high-level scenic details (reservoir, lush bushes, reflections, sky), which readily map onto transcript PIUs such as "pond" and "plants", yielding strong HAR matches and a F1 score of 0.98. However, in the reverse direction, transcript PIUs like "water flowing abundantly" are absent from the model caption and not a direct inference from the image, leading to failed verification in MAR and, hence, an F1 score of 0.25. While high HAR Omission is a universal challenge in these dense scenes, the concurrent spike in HAR Hallucination is model-dependent. This is most evident with Gemma, which exhibits rather high hallucination and omission rates in the Commercial category. An example to substantiate this could be found here Figure 17 (Appendix A.5).

### 4.3 HUMAN JUDGMENT OF CAPTION QUALITY

To further understand human preferences, we conducted a human evaluation study on caption coverage and comprehensiveness. We recruited human participants to rate the captions

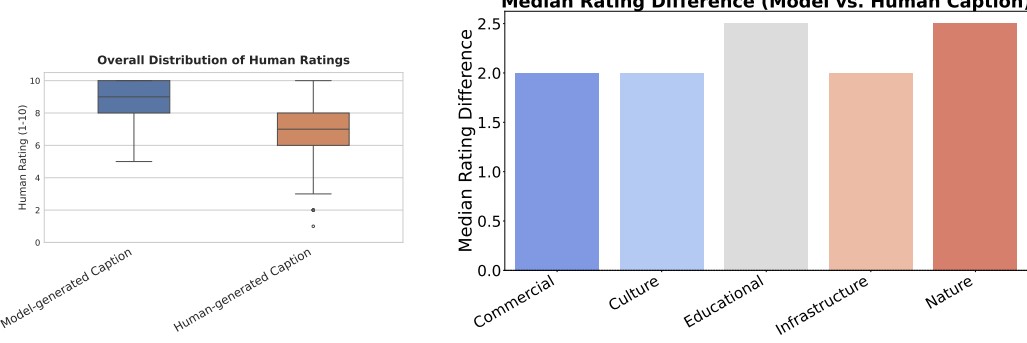

Figure 6: Figure (a) box-plot visualization of comparative human rating for model and human-generated captions and (b) median difference in human ratings between model-generated (Gemini-2.5-Pro) and human-generated captions across five semantic categories.

(human- and model-generated) on a scale of 1–10. The participants were instructed to rate based on how well the captions covered visible details and excluded irrelevant information, focusing on details rather than linguistic fluency. To ensure fair comparison, model-generated captions were constrained to match the median length of the human-generated captions. We randomly selected 22 images, each paired with a human caption and a Gemini-2.5-Pro caption. The ordering of human and model caption were randomized across the images. In this study, 16 native Hindi speakers independently rated all captions.

The results are plotted in Figure 6. The key takeaways are:

1. **Model captions are more preferred:** Figure 6(a) shows the rating distribution of model and human-generated captions. This reflects the greater descriptive consistency of Gemini-2.5-Pro, indicating that the captions capture comprehensive details while presenting them in a more fluent manner. The reduced preference to human captions also mirrors the higher F1 values of HAR over the MAR scores (Figure 3).

2. **Certain categories accentuate the human-model captioning differences:** In Figure 6(b), we delve into the preferential pattern seen in Figure 6(a), by plotting the median difference between the ratings given by the human subjects to the model-generated captions and human-generated captions across the 5 semantic categories. The performance difference is the most pronounced for the 'Nature' and 'Educational' categories, a result that corroborates our category analysis using DCScore (Sec. 4.2.1). The human-ratings reinforce the choice of the DCScore as the tool for the analysis reported in this work.

Although evaluators were instructed to judge content rather than linguistic fluency, several subjects mentioned that human captions at times appeared chopped, grammatically inconsistent or included slang variations. In contrast, model captions maintained a more polished and structured style, which likely contributed to their higher consistency scores.

## 5 Limitations

Our analysis primarily focused on understanding the alignment between natural, informal human captions and model-generated captions in the Vaani dataset, quantified through omission and hallucination rates within a bidirectional evaluation framework. We have also explored only one objective metric - the DCScore for this large-scale study. While the human-ratings provide some justification for this choice, we have not performed comparison with other image captioning metrics.

It is important to note that models have the advantage of generating captions with a length comparable to those produced collectively by multiple humans, whereas each human caption is influenced by the need to caption multiple images in a sequential manner. Although there was no explicit push to speed up the human captioning process, the implicit behavior to caption as many images, may have influenced the human behavior of omitting details of the image. Further, the analysis done in this work is the collective human caption of the single image performed by multiple speakers (on the average, about 1.77 speakers per image). Hence, the combined caption may not follow the natural ordering of the PIUs in the image, a potential factor for the low human rating scores.

## 6 Conclusion

We present a bidirectional evaluation framework that jointly assesses human- and model-generated captions across four state-of-the-art MLLMs. Our study uncovers a fundamental asymmetry: humans exhibit selective attention, describing only the salient aspects of images and often omitting less critical details, whereas models tend to provide more exhaustive coverage. Among the MLLMs, Gemini-2.5-Pro generates the most comprehensive captions with minimal hallucinations or omissions, while GPT-4o demonstrates the most human-like profile, producing fewer hallucinations and having specific descriptions. The human judgments of model- and human-generated captions further strengthen the analysis and findings from the objective metric.

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

# A  APPENDIX

## A.1  BIDIRECTIONAL DCSCORE EVALUATION PIPELINE

Following caption generation, these captions and the corresponding human transcripts were processed through a three-stage evaluation pipeline orchestrated by GPT-4.1 [No thinking] as can be seen in Figure 1. First, in the Decomposition stage (Section 1), Both Model caption and reference caption are broken into Primitive Information Units (PIUs) as per the prompt in Appendix A.2. Next, the Matching stage (Section 2) uses the prompt in Appendix A.2 to map each model PIU to a reference caption PIU ID or "None" if no correspondence is found. In parallel, the Verification stage (Section 3) assesses the factual correctness of each model PIU against the source image or the reference caption, yielding a binary score of "1" for correct and "0" for incorrect (see Appendix A.2). The entire HAR-MAR evaluation process with the intermediate results is illustrated in Figure 2(HAR)(main text) and Figure 7(MAR).

For the Model-as-Reference (MAR) evaluation, we maintain methodological consistency by using the same set of prompts as in the HAR pipeline. This is achieved by simply reversing the inputs: the human transcript is treated as the 'predicted caption' to be evaluated, while the corresponding model-generated caption serves as the 'reference caption' ground truth. As illustrated in our example figures (Figure 2(Main text) and 7), this role reversal is seamless, as the set of PIUs extracted from each text remains identical regardless of its role in the evaluation pipeline.

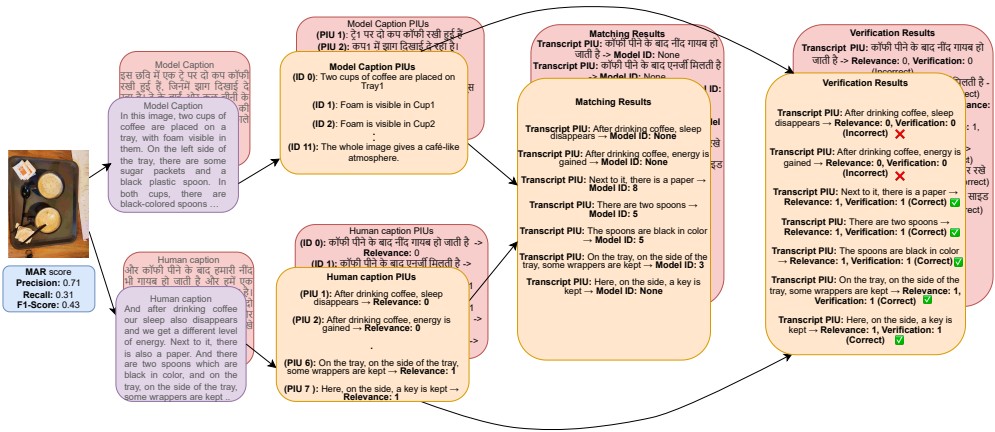

Figure 7: Illustration of Model-as-Reference (MAR) evaluation for a single data point

To fairly handle many-to-one mappings where multiple predicted PIUs may correspond to a single Transcript PIU-DCScore employs a weighting scheme. The contribution of each correct, predicted PIU to the recall score is weighted by $1/N$, where $N$ is the count of prediction PIUs mapped to a single Human-generated caption PIU.

Captions generated from human descriptions of images exhibit substantial variability in both content and level of detail. Some captions are highly elaborate incorporating contextual information, others are minimal, strictly grounded in the visible scene. Such variability introduces inconsistencies, with descriptions that may be underspecified or include information not directly supported by the image. For example, a human caption might describe construction workers as "working very hard" or "They have been working for a few days, and the work has just begun," as shown in Figure 12. These are subjective or temporal assessments rather than literal observations grounded in the image. Consequently, while this reduces precision in the MAR direction – since not all human-generated PIUs can be verified against model captions/image – we still observe a relatively higher MAR Recall, hence a higher F1 score in this case. These deviations show why human captions cannot always be treated as flawless gold references. Since human-generated captions are not completely relaible, we generate machine captions using MLLMs for images in the Vaani dataset and propose a bidirectional human–model evaluation framework that compares human- and model-generated captions to assess the credibility of human annotations.

## A.2 PROMPTS USED IN THE DCSCORE FRAMEWORK

---

### Prompt 1 for Decomposition into PIUs

You are a linguistic expert in extracting primitive information units in the given image caption which is in HINDI and related to the image. Consider the image content when extracting PIUs. In specific, "primitive information units" refer to the smallest standalone pieces of information that collectively represent the entire meaning of the sentence without losing any detail, which typically describe various properties of the visual elements in an image. The primitive information unit should only contain ONE primary element. When extracting primitive information units from image caption, it is useful to assign unique identifiers to the primary objects or entities being discussed. This will help in maintaining clarity and preventing confusion, especially when there are multiple similar objects or entities. For example, if the caption mentions two cats, you can assign unique identifiers such as "cat1" and "cat2" to distinguish them. Besides, for each attribute, you should also assign the identifier to the object it belongs to. Meanwhile, for spatial relationships, you can assign the identifier to the object that is the subject of the relationship in the primitive information unit. For each primitive information unit, you should also need to justify whether the primitive information unit directly describe the image or not. IMPORTANT: Please extract ALL of the primitive information units in the image caption. DO NOT omit any information! Please make sure the identifiers are uniform and only in hindi , with maybe the attached 1,2,3.. like मेज1, फूल1, चटनी1. The output should be a list of dict ["fact": [PRIMITIVE INFORMATION UNIT], "identifier": [UNIQUE ID], "relevance": 1/0, ...] The output of the PIUs should only be in HINDI strictly as input. dont need any justification for the answers just in the explained format.
>>> Caption:

---

### Prompt 2 for matching

You are now a visual-linguistic expert in matching two set of primitive information units generated from A caption which is VLM generated and another Transcript(Human). You will be received a set of predicted VLM primitive information units across a variety of categories and a set of Transcript primitive information units (ground truth). The set of primitive information units is represented as a

---

list of dict ["fact": [PRIMITIVE INFORMATION UNIT], "identifier": [UNIQUE ID], ...] within JSON format. In addition, each primitive information unit in the Transcript set would be accompanied with a unique "id" to identify the Transcript primitive information unit. To match primitive information units (PIUs) from a predicted VLM set with a Transcript set:

1. Preliminary Review: Conduct an initial review of both sets of primitive information units, considering all primitive information units. Understand the details and context presented within each primitive information unit.

2. Inferring Identifier Mappings: Closely examine both sets to deduce potential correlations and mappings based on the content of the primitive information units. Determine if there are any unique identifiers or descriptors that hint at matching entities between the sets. For example, "cat0" in the predicted VLM set's primitive information units may be mapped to "cat1" in the Transcript set's primitive information units. Consider the attribute and spatial relation in both sets for possible mapping.

Please note that there might be some attribute and spatial errors when mapping the objects. Try find the most similar mapping if exists (not need exact matching). If no Transcript primitive information unit matches, simply set matched Transcript id to "None". **IMPORTANT**: Please consider each primitive information unit in the set individually, and MUST NOT omit any primitive information units from the predicted VLM set. You should only output the matching results which will be formatted as a list of dict as ["fact": [PRIMITIVE INFORMATION UNIT],
"identifier": [UNIQUE ID],
"matched_transcript_id": [CORRESPONDING Transcript ID], ...] in JSON format. The "identifier" would be optional, if the item in the fact has already been identified with ids as illustrated in the predicted VLM primitive information units. For key named "matched_transcript_id", the value of "matched_transcript_id" should be the corresponding "index" of the primitive information unit in the Transcript set. For the primitive information unit in the predicted VLM set which cannot be matched with any Transcript primitive information unit, set the value of "matched_transcript_id" to "None". You must produce an output for each VLM predicted primitive information unit, attempting to match it against the transcript set.
>>> Set of VLM predicted Primitive information units:
>>> Transcript Set of Primitive information units:
>>> Matching Result:

---

**Prompt 3 for verification**

You are an extraordinary visual-linguistic expert in verifying the correctness of a set of primitive information units given the image and the corresponding reference caption. The set of primitive information units are extracted from a paragraph of machine-generated image caption of that image. The set of primitive information units is represented as a list of dict ["fact": [PRIMITIVE INFORMATION UNIT], "identifier": [UNIQUE ID], ...] within JSON format. The identifier is unique and to identify the primary objects or entities being discussed. This will help in maintaining clarity and preventing confusion, especially when there are multiple similar objects or entities. For example, if the caption mentions two cats, we would assign unique identifiers such as "cat1" and "cat2" to distinguish them. Besides, for each attribute, it also assigned the identifier to the object it belongs to. Meanwhile, for spatial relationships, it assigned the identifier to the object that is the subject of the relationship in the primitive information unit. You should first go through all of the primitive information units, and understand the details and context presented within each primitive information unit. Then you need to verify the correctness of each individual primitive information units by asking yourself: Statement: "[PRIM-

ITIVE INFORMATION UNIT]" Does the statement correct according to image or reference caption? The output for the predicted VLM set of primitive information units should be formatted as a list of dict as ["fact": [PRIMITIVE INFORMATION UNIT], "identifier": [UNIQUE ID], "verification": 1/0, ...] in JSON format, where 1 represents the fact is correct and 0 represents the fact is incorrect. Other keys in the dictionary are the same as the input. The "identifier" would be optional, if the item in the fact has already been identified with ids as illustrated in the input. Output only the json in the given format, no hallucinations or explanations or justification.
>>> Reference Caption:
>>> Primitive Information Units:

## A.3 Category classification

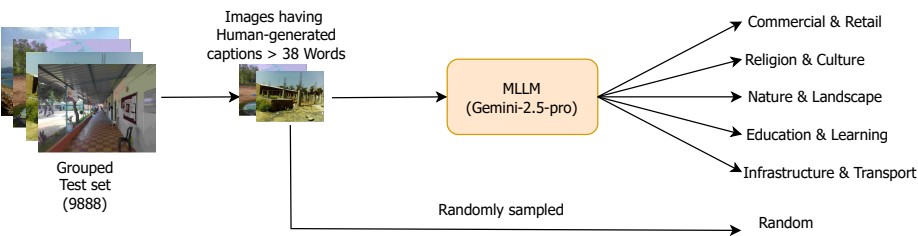

Figure 8: The Pipeline illustrating the selection of images across different categories. we had to first do a multi-fold passes of 1000s of images, This process revealed the five primary categories detailed. We selected only those that were unambiguously classified into a single semantic category

---

**Prompt used for category classification through Gemini-2.5-Pro**

You are an expert image classifier with a deep understanding of Indian cultural contexts. Your task is to analyze an image and classify it based on its relevance to the categories defined below. For each image, identify the most appropriate category. In most cases, there should be a single dominant category that captures the main theme of the image. Only in rare cases—when two categories are both clearly represented—may you assign two categories. Even then, only one can be rated as High (the central theme), while the other must be rated as Medium or Low. For every category assigned, provide a rating (High, Medium, or Low) based on its prominence in the image, along with a brief justification. — Categories & Definitions

1. Commercial & Retail: Scenes centered on buying, selling, or trade. Examples: Markets, street vendors, kirana stores, malls, commercial signage. 2. Religion & Culture: images showcasing religious practices, cultural heritage, or festivals. Examples: Temples, mosques, church, pujas, traditional attire (saree, kurta), cultural performances. 3. Nature & Landscape: Outdoor scenes dominated by natural elements. Examples: Mountains, rivers, beaches, forests, gardens, wildlife. 4. Education & Learning: Settings related to academic or instructional activities. Examples: Schools, classrooms, students in uniform, libraries, teachers. 5. Infrastructure & Transport: Man-made public works and modes of transit. Examples: Roads, bridges, railway stations, airports, buses, trains, auto-rickshaws. —

Rating Guidelines When assigning a rating (High, Medium, Low) to a category, carefully consider the intensity and dominance of that theme in the image: 1. High → The category is the clear central theme of the image, visually dominant and unambiguous. 2. Medium → The category is present and noticeable, but not the primary focus. 3. Low → The category is only weakly or marginally represented, secondary to other themes. Do not default to "High" whenever a category is present. Reserve High only for strong, central cases. Use Medium or Low where the category is visible but less dominant. —

Output Format Respond with a JSON array of objects. Each object represents a classified category and must include category, rating, and reasoning. The array will contain one object, or in cases of significant overlap, two objects.
- If the image does not contain significant elements from any category, classify it as "Normal" with a rating of "N/A". - Do not make assumptions or hallucinate; categorize based only on visual evidence.
Example Output (Single Category):

```
[
  {
    "category": "Infrastructure & Transport",
    "rating": "High",
    "reasoning": "The image is a wide shot of a bustling railway station,
    with trains and passengers as the central focus."
  }
]
```

Example Output (Multiple Categories):

```
[
  {
    "category": "Religion & Culture",
    "rating": "High",
    "reasoning": "The central subject is a group of people in vibrant,
    traditional attire participating in a religious procession."
  },
  {
    "category": "Commercial & Retail",
    "rating": "Medium",
    "reasoning": "The procession is taking place in a crowded market
    street, with numerous shops and street vendors clearly visible
    in the background."
  }
]
```

### A.4 CATEGORY EXAMPLES

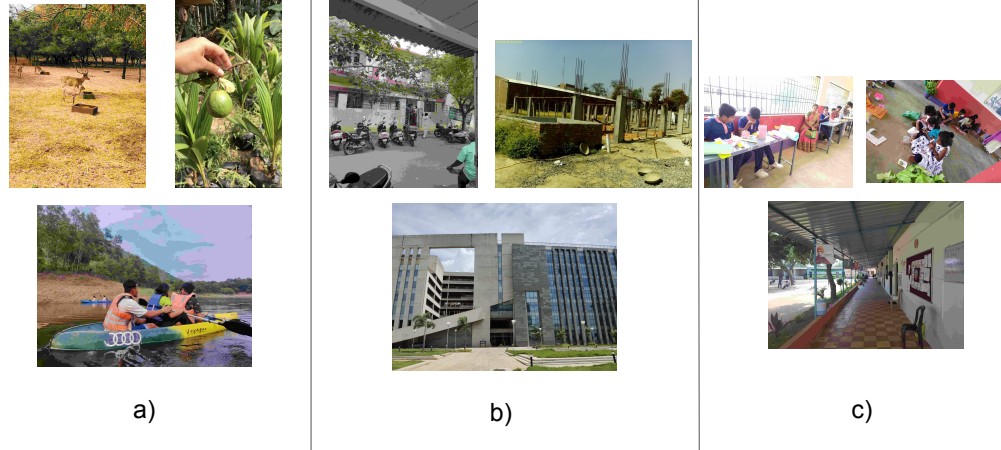

a)                          b)                          c)

Figure 9: image examples from a) Nature & Landscape, b) Infrastructure & Transport and c) Education & Learning categories during sub-categorization.

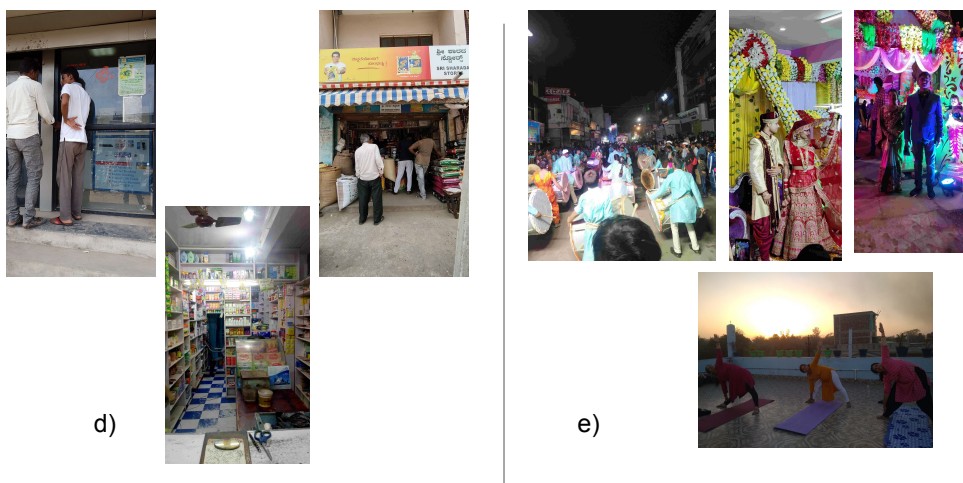

Figure 10: image examples from d) Commercial & Retail, e) Religion & Culture categories during sub-categorisation.

## A.5   Additional Results

### A.5.1   Word-Length-Constrained Model Performance Analysis

In parallel to the word-length constraints put on model captions for human judgment, we also examine how trends and variations occur when model captions are constrained to the median length of human-generated captions.We would also like to point out further that the aggregation of multiple speaker captions for a single image does not yield the strong overlap one might intuitively expect—in fact, the overlap is considerably limited. For example, if 5 people described an image but only one mentioned a given PIU, its consensus is 1. If two people mentioned it, the consensus is 2, and so on. These PIU-level consensus counts were averaged at the image level and subsequently aggregated by category. The distributions shown in the figure 11b indicate that consensus across speakers is generally low, hovering near 1.25, highlighting that even when multiple individuals describe the same image, In most cases they tend to focus on different subjects. Notably, Nature images exhibit somewhat higher variability in comparison to other categories which reflect limited overlap. This effectively does away with the argument of aggregation of the same subjects over many speakers.

Looking at model-wise comparisons on Random images under these constraints, we see the same overall trends as in the unconstrained setting, though the differences across models are less pronounced, likely owing to smaller variations in caption length. The performance difference in both HAR and MAR direction for Gemini-2.5-Pro could be found in figure 11a. In the HAR direction, Gemini-2.5 Pro continues to lead, followed by GPT-4o and Gemma-3-12B. In the MAR direction, GPT-4o maintains an edge, while Gemini and Gemma cluster lower. Notably, LLaMA has been excluded from the constrained caption plots due to its gross under performance, with a majority of outputs exhibiting hallucinations as can be seen in the Hallucination and Omission rates plots in Figure 18. Gemma-3-12B continues to exhibit the highest hallucination rate among the competitive models. The Human omission rates (MAR) also remain significantly higher than the Model omission rates (HAR), solidifying the claim that humans demonstrate strong visual selectivity. Among the models, LLama-4-17B displays the highest hallucination rate in the constrained captioning setting, underscoring its lack of stability under these conditions.

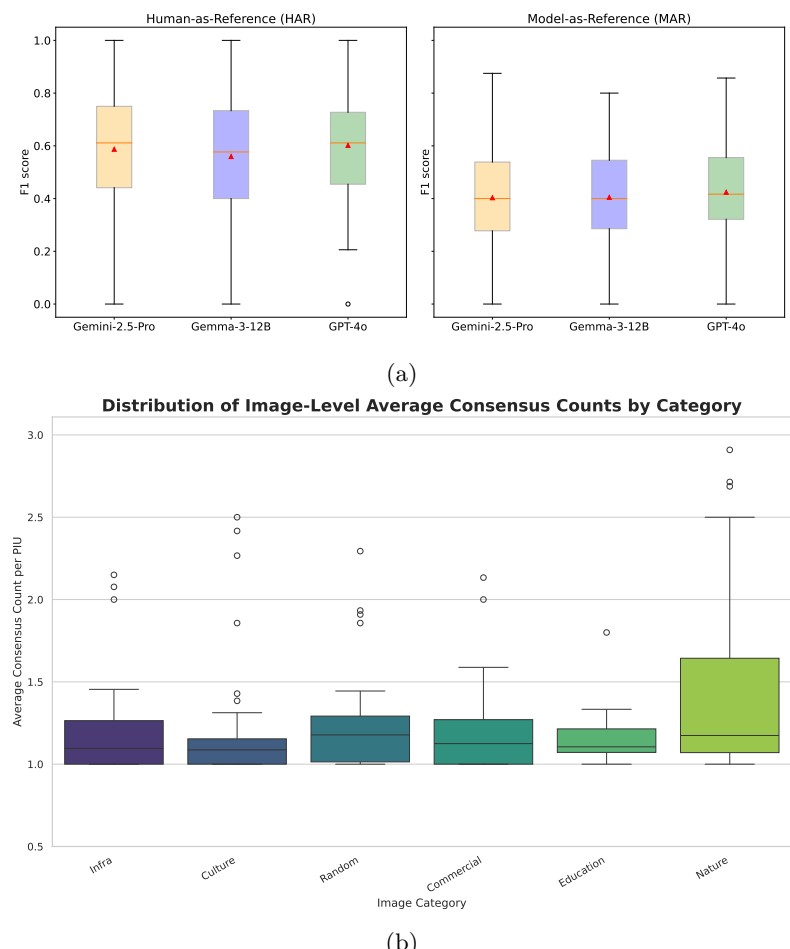

Figure 11: Figure (a) Depicts a Boxplot vizualisation of the DCScore (HAR and MAR) of Random images with in length constrained caption setting across three models(Trends stay consistent) and Figure (b) shows category-wise Distribution of PIU consensus scores, showing limited descriptive overlap among human annotators

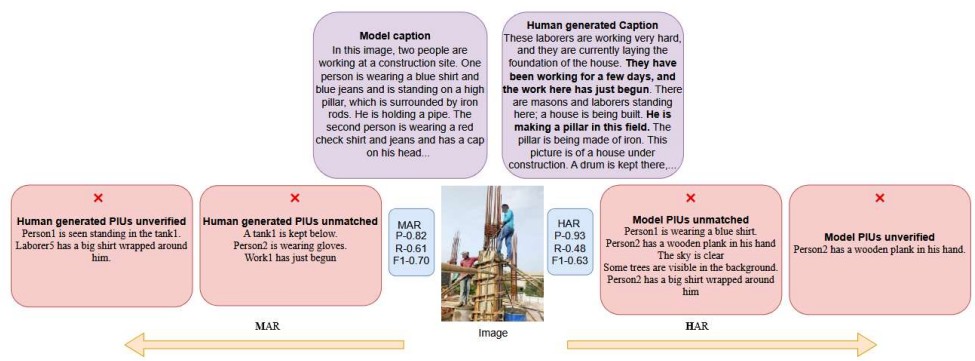

Figure 12: Example of bidirectional evaluation highlighting differences between human- and model-generated captions.

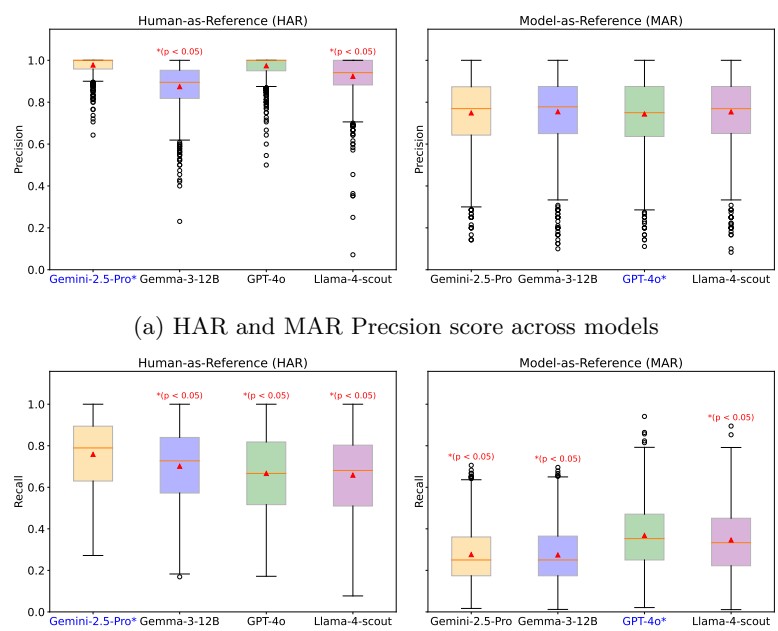

(a) HAR and MAR Precsion score across models

(b) HAR and MAR Recall score across models

Figure 13: Precison and Recall score for Random category across models

### A.5.2 HAR and MAR evaluations

Figure 12 illustrates an example of bidirectional evaluation highlighting differences between human- and model-generated captions. The human caption in this example includes narrative elements such as "they have been working for a few days" that are not directly grounded in the visual content. This reflects a broader trend in which humans often elaborate beyond visible cues, drawing on personal experience or cultural associations. By contrast, the model-generated caption (GPT-4o) introduces an error by incorrectly identifying a wooden plank in Person 2's hand, yet it provides a more exhaustive description of the observable scene. Such differences emphasize the asymmetry between human and model captions - humans enrich descriptions with context, while models prioritize coverage of the visible scene.

In addition to the F1 score results reported in the main text (Section 4.1), we provide a detailed analysis of precision and recall for both Human-as-Reference (HAR) and Model-as-Reference (MAR) evaluations in Figures 13a and 13b, respectively. These figures show precision and recall distributions across models and highlight the trade-offs between exhaustive coverage (high recall) and alignment with reference content (high precision). The additional metrics help contextualize the DCSCORE differences observed in the main text and illustrate model-specific tendencies toward exhaustive coverage and reduced selectivity.

Figure 20 shows the boxplots of F1 scores across five categories - Commercial, Infrastructure, Nature, Culture, Education - as well as Random, for the four evaluated models under both HAR and MAR settings.

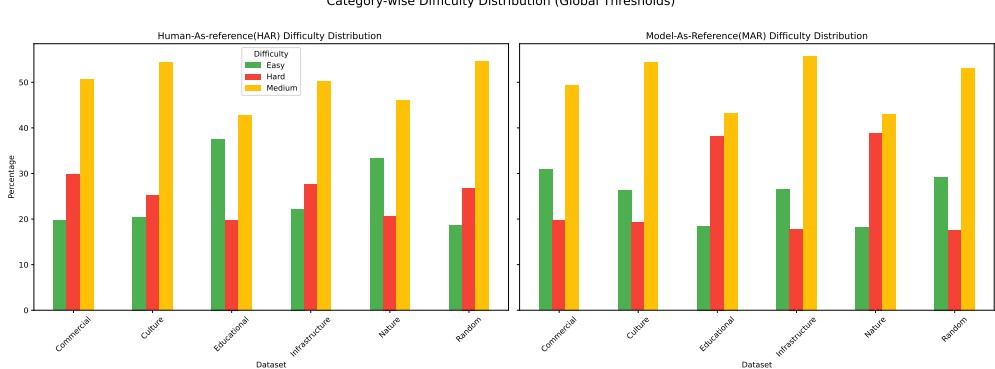

Figure 14: Difficulty distribution of images across categories in both HAR/MAR (F1) setting. Nature and Education categories contain the highest proportion of 'Easy' samples in HAR setting.

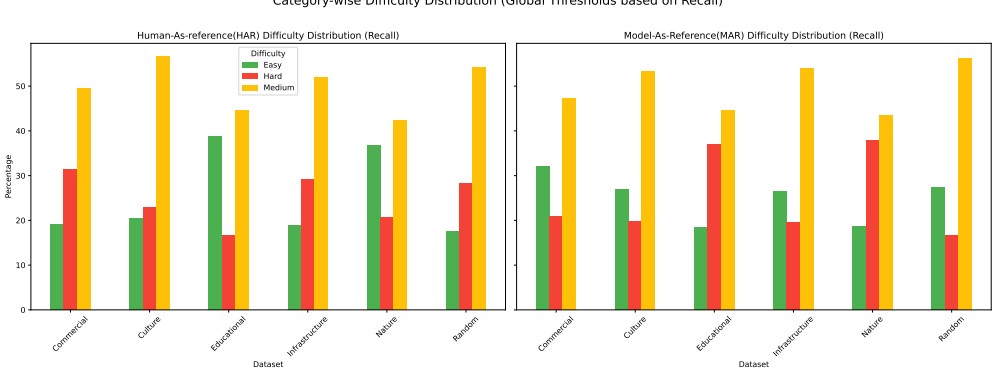

Figure 15: Difficulty distribution of images across categories in both HAR/MAR (Recall) settings. Nature and Education categories contain the highest proportion of 'Easy' samples in HAR setting.

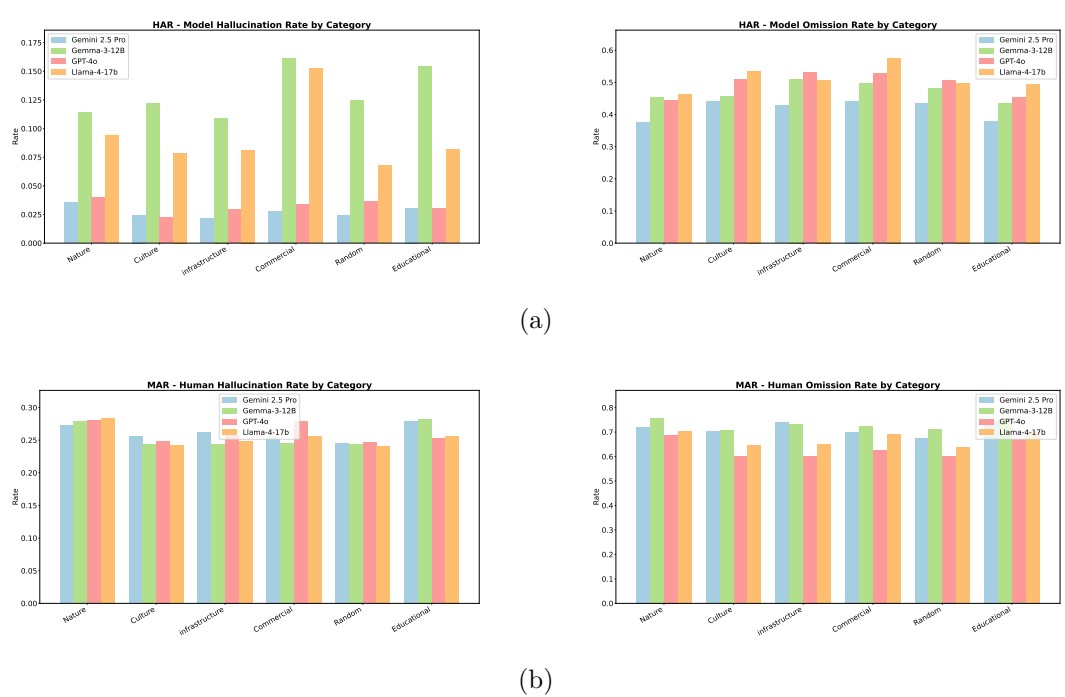

(a)

(b)

Figure 16: Figure (a) Depicts Hallucination and Omission rates across categories in HAR setting and (b) Hallucination and Omission rates across categories in MAR setting, with Gemma showing the highest hallucination across categories.

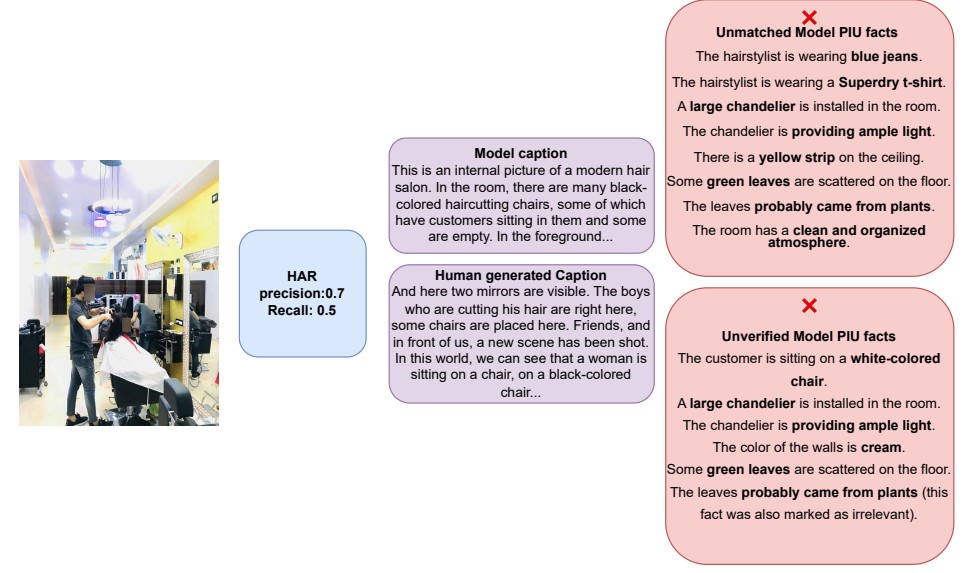

Figure 17: Example of a Commercial category image for which **Gemma** generated caption with Low HAR precision and recall. In contrast, the Gemini model generated caption boasts precision of 0.96 and Recall of 0.68.

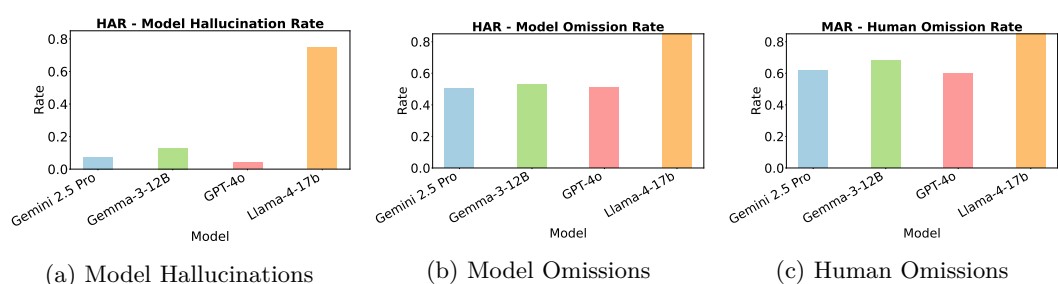

(a) Model Hallucinations     (b) Model Omissions     (c) Human Omissions

Figure 18: Bidirectional hallucination and omission rates for word-length constrained model captions: (a) Model hallucinations in the HAR setting, (b) Model omissions in the HAR setting, and (c) Human omissions when model captions serve as reference (MAR)

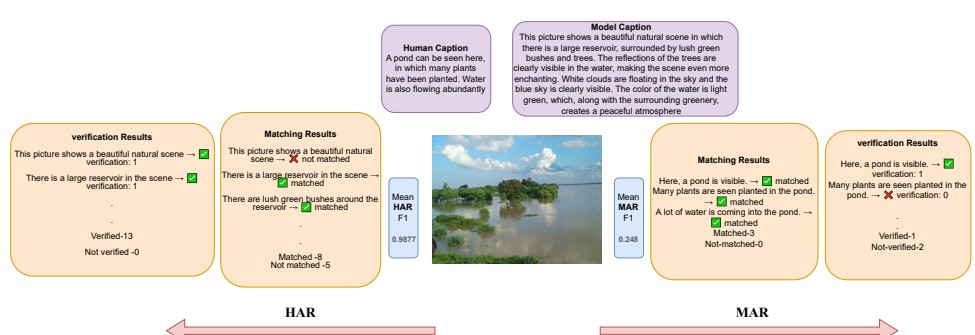

Figure 19: Example of an image in Nature, where it is classified as 'Easy' in HAR but 'Hard' in MAR.

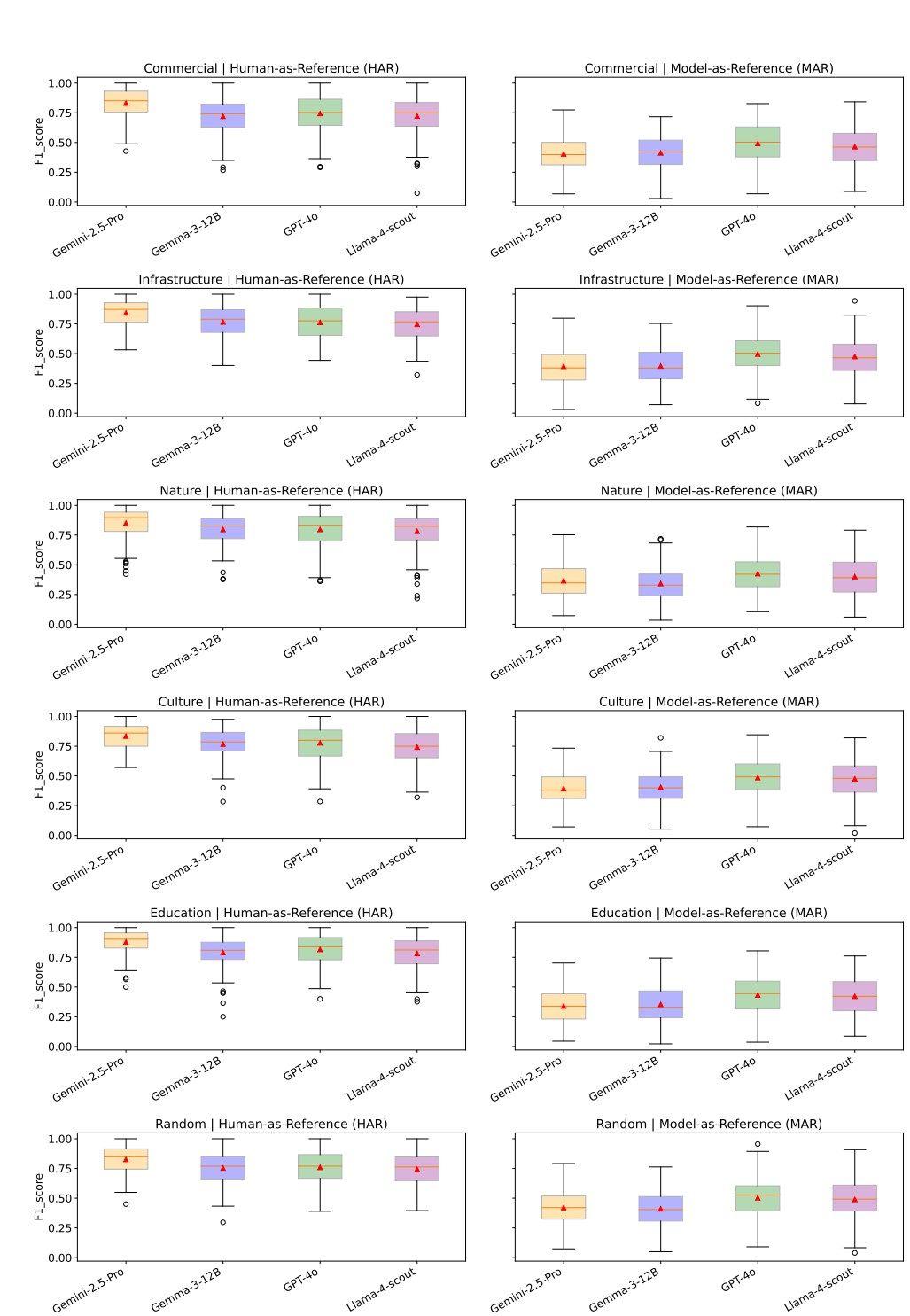

Figure 20: F1 score across categories - Commercial, Infrastructure, Nature, Culture, Education and Random for the four MLLMs under HAR and MAR settings.

