# OpenReview forum: "See both ways: A bidirectional evaluation of Multimodal Language Models and Human Spontaneous Speech for Image Captioning"
_ICLR.cc/2026/Conference — ICLR 2026 Conference Withdrawn Submission_

### Official Review · Reviewer_u9Nu · 2025-10-28

**Soundness:** 2
**Presentation:** 3
**Contribution:** 1
**Rating:** 2
**Confidence:** 3

**Summary:**

This paper centers on the visually-grounded speech dataset, Vaani.
This dataset consists of images and associated spoken captions collected from people in India speaking various Indic languages.
The paper compares the human captions to captions generated by four vision-language models.
As a primary analysis tool, the authors propose applying the DCScore metric bidirectionally (using humans as reference and models as reference).
The main conclusions reveal that spoken captions are shorter, more casual, and incomplete, whereas model outputs provide more complete descriptions; notably, the latter are also preferred by human raters.

**Strengths:**

- I am sympathetic to the general task of working with realistic visually-grounded speech datasets, and I applaud the effort to work with the challenging Vaani data. This type of data holds significant potential for developing technology for low-resource languages (such as Indic languages).
- The human evaluation study (Section 4.3) is valuable. Beyond validating the results, the collected ratings could serve as a useful resource for identifying or filtering unreliable human captions within the Vaani dataset, thereby improving its utility for downstream multimodal learning tasks.

**Weaknesses:**

- The paper claims to compare VLMs with human captioning, but this comparison is conducted only in a very specific setting on a single dataset. It is unclear whether the conclusions would generalize to other datasets. For example, as the authors observe, "several subjects mentioned that human captions at times appeared chopped, grammatically inconsistent or included slang variations." I believe this reflects challenges inherent to this dataset (e.g., data collection in the wild, demographics) rather than general characteristics of spoken captions. Note that other spontaneously spoken caption datasets exist, such as Places Audio Captions (Harwath et al., 2016) and its extensions to Hindi (Harwath et al., 2018) and Japanese (Ohishi et al., 2020).
- Another central claim of the paper is the introduction of a new evaluation framework (the bidirectional DCScore). But this contribution is never compared to existing metrics (BLEU, ROUGE, CIDEr, METEOR, or other model-based metrics). Do we really need this new machinery? Would we reach similar conclusions using other metrics?
- All in all, I think the true contribution of the paper is an analysis of the Vaani dataset. The insights are valuable, but narrow in scope, as the conclusions rely heavily on the specific characteristics of this dataset.

References:
- Harwath, David, Antonio Torralba, and James Glass. "Unsupervised learning of spoken language with visual context." _NeurIPS_, 2016.
- Harwath, David, Galen Chuang, and James Glass. "Vision as an interlingua: Learning multilingual semantic embeddings of untranscribed speech." _ICASSP_, 2018.
- Ohishi, Yasunori, et al. "Trilingual semantic embeddings of visually grounded speech with self-attention mechanisms." _ICASSP_, 2020.

**Questions:**

- As I understand it, the evaluation is conducted in Hindi. Could it also be performed in English? That is, could the human captions be translated from Hindi to English and have the VLMs output in English? Would the results look similar?
- Related to the above: Doesn't working in Hindi (a less-resourced language than English) introduce errors in the DCScore pipeline (decomposition, matching, verification)?
- In Figure 4, I would be curious to see the human hallucination plots. Are those results as low as the model hallucinations?
- The paper might benefit from a discussion of the literature on spoken versus written image descriptions (e.g., van Miltenburg et al., 2018), which explores how modality can influence descriptive style (adverb use, hedging, verbosity).
- The paper might also benefit from discussing the literature on modeling what humans tend to mention in images (e.g., Berg et al., 2012), which examines how humans prefer certain image elements when describing scenes (although this work focuses on written rather than spoken captions).

References:
- Berg, Alexander C., et al. "Understanding and predicting importance in images." _CVPR_, 2012.
- Van Miltenburg, Emiel, Ruud Koolen, and Emiel Krahmer. "Varying image description tasks: spoken versus written descriptions." _Workshop on NLP for Similar Languages, Varieties and Dialects_. 2018.

---

> ### Author Response · Authors · 2025-12-03
>
> **Addressing the weaknesses:**
> > **1)** Based on extension to Places Audio Captions dataset
>
> We thank the reviewer for highlighting the Places Audio Captions dataset [1]. To address this, we conducted a new experiment using the Hindi extension of the dataset.
>
> **Experimental Setup:** We evaluated a random subset of 200 images with their corresponding Hindi spoken transcripts using Gemini-2.5-Pro and Gemma-12B-it, maintaining the exact prompt settings used for the Vaani dataset.
>
> The results strongly corroborate the trends observed in our main paper, confirming that the characteristics we reported are inherent to spoken image descriptions, not specific to the Vaani dataset.
>
> - **Human-As-Reference (HAR) Remains Robust:** Consistent with our main results, we observe that the HAR scores are significantly higher than the MAR scores. Furthermore, Gemini-2.5-Pro exhibits a lower model omission rate on this dataset compared to Vaani (0.28 vs 0.43), and its model hallucination rate also remains low (0.037), comparable to the Vaani dataset (0.024).
>
> - **Human Selectivity is Even More Pronounced (MAR):** The trend of high human selectivity observed in Vaani is even more pronounced in the Places Audio Captions dataset with sparse and less dense captions; the Human Omission Rate spiked to 0.851, compared to 0.673 in Vaani (Gemini-2.5-pro). Hence, the fact that our proposed metric trends hold and are even accentuated on this dataset provides strong evidence for the generalizability of our conclusions. [check for HAR/MAR boxplots on https://ibb.co/8nt9cM62]
>
> [1] D. Harwath, G. Chuang, and J. Glass, "Vision as an interlingua: Learning multilingual semantic embeddings of untranscribed speech," Proc. ICASSP, Calgary, Canada, April 2018
>
> > **2)** Another central claim of the paper is the introduction of a new evaluation framework (the bidirectional DCScore). But this contribution is never compared to existing metrics (BLEU, ROUGE, CIDEr, METEOR, or other model-based metrics). Do we really need this new machinery? Would we reach similar conclusions using other metrics?
>
> To determine if existing metrics could suffice, we computed BLEU-4, METEOR, ROUGE-L, and BERTScore for our dataset and analyzed their Spearman correlation with the human judgments collected in Section 4.3.
>
>
> | Metric | Spearman Correlation (ρ) (Model-As-reference/MAR) | Spearman Correlation (ρ) (Human-As-reference/HAR) |
> | :--- | :--- | :--- |
> | **Bidirectional-HAR/MAR (Ours)** | **0.263** | **0.121** |
> | BERTScore | 0.206 | -0.065 |
> | METEOR | 0.165 | -0.058 |
> | ROUGE-L | 0.156 | -0.002 |
> | BLEU-4 | -0.075 | -0.075 |
>
>
> **Key Findings:**  As shown in the table below, our proposed metric (Bidirectional-HAR/MAR) is the only method that aligns closest positively with human judgment in both directions. Relying on the other metrics mentioned would be statistically misaligned with the human judgements recorded, necessitating the proposed bidirectional framework.
>
>
> **Analysis:**
> - In the Human-as-Reference setting, all traditional metrics exhibit negative correlations (e.g., BLEU-4 at -0.075, BERTScore at -0.066). This confirms that lexical and embedding-based matching fail when evaluating spontaneous, speech . They penalize the valid but structurally colloquial human transcripts that our PIU-based approach correctly handles.
>
> - In the Model-as-Reference setting (where the caption is polished), our MAR score (0.264) still outperforms the strongest baseline, BERTScore (0.206). This indicates that evaluating human input requires the granular, fact-based decomposition offered by our pipeline, rather than the broad semantic similarity measured by embeddings.

---

> ### Author Response · Authors · 2025-12-03
>
> **Questions:**
> >**Q1) and Q2)**
> As I understand it, the evaluation is conducted in Hindi. Could it also be performed in English? That is, could the human captions be translated from Hindi to English and have the VLMs output in English? Would the results look similar?
> Related to the above: Doesn't working in Hindi (a less-resourced language than English) introduce errors in the DCScore pipeline (decomposition, matching, verification)?
>
> To address the questions regarding language dependency, we conducted a full evaluation by translating the dataset and repeating the intermediary pipeline steps in English. We used an LLM-based clause-to-clause translation for human transcripts to preserve the original syntactic structure and 'spontaneous' nature of the speech. As shown in the table below, the primary rankings remain consistent.
> | Model   | HAR (Hindi) | HAR(English) | Δ (Delta) | MAR(Hindi) | MAR(English) | Δ (Delta) |
> |---------|------------------------|--------------------------|-----------|--------------------------|----------------------------|-----------|
> | **Gemini** | **0.8263** | **0.8354** | **+0.0091** | 0.4212 | 0.3530 | -0.0681 |
> | **Gemma**  | 0.7536 | 0.7516 | -0.0021 | 0.4109 | 0.3651 | -0.0458 |
> | **GPT-4o** | 0.7593 | 0.6816 | -0.0777 | **0.5032** | **0.6168** | **+0.1136** |
> | **Llama**  | 0.7430 | 0.7744 | +0.0031 | 0.4894 | 0.3923 | -0.0970 |
>
> Gemini-2.5-Pro remains the statistically significant leader in the HAR (Description) direction, while GPT-4o remains the leader in the MAR (Retrieval) direction. This confirms that our main paper's conclusions are not artifacts of the Hindi pipeline.
>
> - Gemini-2.5-Pro shows stability in HAR , indicating that its factual grounding capabilities are equally robust in both Hindi and English. In the MAR setting, GPT-4o sees a significant performance boost in English.
>
> - This might suggest that GPT-4o's English captions align much more closely with human descriptive patterns than its Hindi captions. Crucially, the leaderboard remains unchanged, validating the robustness of our original analysis.
>
> > **Q3** In Figure 4, I would be curious to see the human hallucination plots. Are those results as low as the model hallucinations?
>
> Please find the Human Hallucination(Imagination/over-generation) plots in here: [https://ibb.co/Z6Q298cW]. No, the Human Hallucination is higher than the model hallucination, possibly because humans frequently mention specific, subjective, or culturally dense details that the models acting as reference fail to contain.
>
> > **Q4)** and **Q5)**
> The paper might benefit from a discussion of the literature on spoken versus written image descriptions (e.g., van Miltenburg et al., 2018), which explores how modality can influence descriptive style (adverb use, hedging, verbosity).
> The paper might also benefit from discussing the literature on modeling what humans tend to mention in images (e.g., Berg et al., 2012), which examines how humans prefer certain image elements when describing scenes (although this work focuses on written rather than spoken captions).
>
> We thank the reviewer for these excellent references. We agree that contextualizing our findings within the broader literature on modality and human salience will significantly strengthen the paper. We will incorporate a dedicated discussion of these works in the final version.

---

### Official Review · Reviewer_3M5F · 2025-10-29

**Soundness:** 3
**Presentation:** 3
**Contribution:** 3
**Rating:** 6
**Confidence:** 3

**Summary:**

This paper presents a novel bidirectional score (HAR and MAR) for image captioning datasets to evaluate the similarity between human-generated and model-generated captions. Rather than English-only captions, this paper explores more of Hindi speech with large-scale Vanni dataset. The experiments on both close-source and open-source models are comprehensive and thorough.

**Strengths:**

1.	The motivation of this paper is interesting.
2.	The introduction of spontaneous and image-prompted speech is novel for image caption evaluation.
3.	Comprehensive experimental results demonstrate the rationality and effectiveness of proposed HAR score and MAR score.

**Weaknesses:**

1.	Only DCScore is used to evaluate the model-generated caption and human-generated caption, while some reference-based metrics BLEU, METEOR, and BERTScore are not exploited.
2.	Multimodal models typically tend to generate complex clauses with more than 50 words given an image, while humans usually deliver simple sentences. Does this phenomenon affect the final score?
3.	Some mistakes in the paper, such as Line 356.
4.	The format of this paper seems different from other ICLR papers.

**Questions:**

Please see the above weaknesses.

---

> ### Author Response · Authors · 2025-12-03
>
> > **Q1)** Only DCScore is used to evaluate the model-generated caption and human-generated caption, while some reference-based metrics BLEU, METEOR, and BERTScore are not exploited.
>
> We appreciate the suggestion to broaden our evaluation. In response, we have computed BLEU-4, METEOR, ROUGE-L, and BERTScore for our dataset and analyzed their alignment with the human judgments collected in Section 4.3.
>
> | Metric | Spearman Correlation (ρ) (Model-As-reference/MAR) | Spearman Correlation (ρ) (Human-As-reference/HAR) |
> | :--- | :--- | :--- |
> | **Bidirectional-HAR/MAR (Ours)** | **0.263** | **0.121** |
> | BERTScore | 0.206 | -0.065 |
> | METEOR | 0.165 | -0.058 |
> | ROUGE-L | 0.156 | -0.002 |
> | BLEU-4 | -0.075 | -0.075 |
>
> **Analysis:**
> - In the Human-as-Reference setting, all traditional metrics exhibit negative correlations (e.g., BLEU-4 at -0.075, BERTScore at -0.066). This confirms that lexical and embedding-based matching fail when evaluating spontaneous, speech . They penalize the valid but structurally colloquial human transcripts that our PIU-based approach correctly handles.
>
> - In the Model-as-Reference setting, our MAR score (0.264) still outperforms the BERTScore (0.206). This indicates that evaluating human input requires the granular, fact-based decomposition offered by our pipeline, rather than the broad semantic similarity measured by embeddings.
>
> > **Q2)** Multimodal models typically tend to generate complex clauses with more than 50 words given an image, while humans usually deliver simple sentences. Does this phenomenon affect the final score?
>
> - We would like to firstly address that the metric is designed to be agnostic to linguistic complexity By decomposing captions into 'Primitive Information Units' (PIUs), the metric evaluates the presence of atomic facts rather than sentence structure. Whether information is conveyed via complex clauses or simple sentences, the PIU extraction isolates the underlying content for matching and verification.
>
> - We also explicitly ruled out verbosity bias and conducted a controlled experiment where model captions were constrained to match the median length of human transcripts. As shown in Figure 11, the performance trends remained consistent with the unconstrained setting: Gemini-2.5-Pro retained the highest HAR score and GPT-4o retained the highest MAR score. We address the impact of caption length and complexity in Appendix A.5.1 and Section 3.2.
>
> > **Q3) and Q4)**
>
> **(based on line error and format)**
>
> We confirm that this manuscript was prepared using the official ICLR 2026 Master Template (retrieved from GitHub) and passed all automated formatting checks during the submission process. We will ensure strict adherence to all style files in the final revision. We further appreciate your careful reading and shall rectify the error (Line 356) in the camera-ready version.

---

### Official Review · Reviewer_Mz2K · 2025-11-01

**Soundness:** 2
**Presentation:** 2
**Contribution:** 2
**Rating:** 4
**Confidence:** 4

**Summary:**

This paper investigates the image captioning capabilities of existing MLLMs from the perspective of human-speech transcription. The authors use the Vaani dataset as the source for human-speech transcription of images and analyze image captions generated by MLLMs such as Gemini 2.5 Pro, GPT-4o, Gemma-3-12B, and Llama-4-scout-17b-16e-instruct. The experiments using the evaluation metrics proposed by a recent study show that humans are more selective than models.

**Strengths:**

1. This paper is clear and well-organized overall.
2. The attempt to compare human spoken transcriptions with model-generated image captions is interesting.

**Weaknesses:**

1. Why do we need to compare human spoken transcriptions with model-generated image captions? What is the motivation of this study?
2. Why do we use precision and recall scores? Are these metrics well aligned with the motivation?
3. Section 3.2 describes how to evaluate generated captions using human transcriptions as references. However, Section 3.3 defines a bidirectional evaluation framework without providing its context. This confuses understanding the following sections.

**Questions:**

(Copied)
1. Why do we need to compare human spoken transcriptions with model-generated image captions? What is the motivation of this study?
2. Why do we use precision and recall scores? Are these metrics well aligned with the motivation?
3. Why do we use a bidirectional evaluation framework?

---

> ### Author Response · Authors · 2025-11-28
>
> > **Q1:** Why do we need to compare human spoken transcriptions with model-generated image captions? What is the motivation of this study?
>
> We undertake this comparison because spontaneous speech represents the most fundamental natural form of 'verbal captioning.' Unlike traditional datasets that rely on carefully curated, text-based annotations, spontaneous speech captures the real-life scenario of human description, where the descriptions are often sparse and linguistically unstructured (Section 3.1 ).The motivation for this study is to quantify the alignment gap between image captioning performed by modern MLLMs and spontaneous, natural human speech-based captioning of an Image.
>
>
> > **Q2:** Why use precision and recall scores? Are they well aligned?
>
> * The effort to use Precision and Recall is completely in tandem with the DCscore paper [1] which proposes these measures to understand differences between a pair of captions. Specifically, these measures serve definitions for the two primary axes of  investigation: Hallucination (factual correctness) and Omission (descriptive coverage). Yes, These metrics are strictly aligned with our motivation to audit the alignment gap between human vs. model captions with the help of an F1 Score (established through the human judgment experiments).
>
> * As we had highlighted in Section 4.1, Precision in Human-As-Reference (HAR) helps measure Hallucination: This allows us to quantify how often models incorrectly add visual details absent in the image (as seen with Gemma-3-12B in Figure 4a).Recall helps measure comprehensiveness and Omission: Together, Precision and Recall are specific tools required to capture the 'fundamental asymmetry' between human communicative sufficiency and model exhaustiveness described in our Conclusion.
>
> [1] Ye, Qinghao, et al. "Painting with words: Elevating detailed image captioning with benchmark and alignment learning." arXiv preprint arXiv:2503.07906 (2025).
>
>
> > **Q3:** Why do we use a bidirectional evaluation framework?
>
> * As stated in Section 3.1, we employ a bidirectional framework in this study as we consider a setting which does not contain a   ‘‘ground-truth’’. Unlike traditional captioning tasks, where annotators are instructed to be comprehensive, the Vaani dataset involves largely diverse, colloquial speakers providing natural, spontaneous captions for an image. The bi-directional framework allows us to compare and contrast the captions in a reference-free framework.
>
> * The Human-as-Reference (HAR) direction primarily identifies Model Hallucinations and Omissions. By verifying unmapped model details against the source image, we ensure models are assessed on correctness without being penalized for exceeding the human's selective description. However, the Model-as-Reference (MAR) direction evaluates Human Selectivity and Imagination. By treating the model caption as the reference, we are able to quantify exactly how much visual information humans spontaneously skip.

---

### Official Review · Reviewer_KAyz · 2025-11-02

**Soundness:** 1
**Presentation:** 2
**Contribution:** 1
**Rating:** 0
**Confidence:** 5

**Summary:**

This manuscript aims to investigate the relationship between model-generated captions and human spontaneous speech descriptions, using the Hindi subset of the VAANI dataset. The authors introduce a “bidirectional” evaluation framework, using Human-as-Reference (HAR) and Model-as-Reference (MAR) metrics, derived from DCScore. The study benchmarks several multimodal large language models (MLLMs) such as Gemini 2.5 Pro, GPT-4o, Gemma-3-12B, and LLaMA-4-scout.

While the topic may be of mild interest to researchers working on multimodal evaluation or cross-lingual captioning, it does not fit within ICLR’s scope. The paper presents no new learning representations, algorithms, or optimization frameworks. Its contributions are descriptive and incremental at best, with no methodological novelty or theoretical advancement.

**Strengths:**

- Provides an empirical analysis of MLLM-generated captions versus spontaneous human speech in Hindi, an underrepresented language.
- The bidirectional evaluation idea is straightforward and may help analyze asymmetries between human and model captions.
- The writing is relatively clear and the experimental setup is well documented.
- The dataset (VAANI) is interesting from a sociolinguistic and inclusivity standpoint.

**Weaknesses:**

- The work does not propose any new learning paradigm, model, or representation. It focuses entirely on empirical evaluation, making it unsuitable for ICLR. The analysis remains within the scope of applied benchmarking rather than representation learning.
- The so-called bidirectional evaluation merely swaps the roles of reference and prediction in an existing metric (DCScore). This is a trivial extension that does not constitute a meaningful methodological or conceptual advance.
- The study provides no analysis of internal representations, cross-modal alignment, or learning dynamics. Thus, it offers no new understanding of how multimodal models learn or encode information.
- The reliance on GPT-4.1 for decomposition, matching, and verification introduces uncontrollable biases and threatens reproducibility.
- There is no verification that GPT-4.1 judgments correlate with human assessments beyond anecdotal claims.
- The human evaluation is too small (22 samples, 16 raters) to support any statistically robust conclusions.
- Statistical tests (Welch t-tests) are misapplied given the small and non-independent samples.
- The use of a single dataset (VAANI Hindi subset) restricts generalization.
- The paper does not justify why DCScore is an appropriate metric for spontaneous speech descriptions.
- There is no comparison to established captioning metrics (CIDEr, SPICE, CLIPScore, BLIPScore), which weakens the methodological soundness.
- The conclusions—e.g., humans are more selective while models are more exhaustive—are well known and uninformative. The analysis reiterates established observations without new theoretical framing or insight.
- The authors draw sweeping claims about “human imagination” and “model verbosity” without sufficient grounding in cognitive or psycholinguistic evidence.

**Questions:**

- How is your proposed bidirectional framework conceptually distinct from computing DCScore in both directions?
- Did you obtain ethical clearance or participant consent for using the VAANI data?
- How do you ensure the reproducibility and fairness of GPT-4.1-based evaluation steps, given its proprietary nature?
- Can your conclusions about “human imagination” and “model selectivity” be supported empirically, or are they speculative?
- Why are standard captioning metrics omitted from comparison?
- How does this work advance representation learning, which is the core theme of ICLR?

**Details Of Ethics Concerns:**

Use of human speech data:
The paper heavily relies on human audio data from the VAANI dataset, yet provides no ethical discussion of data provenance, consent, or participant anonymization. The VAANI dataset includes demographic metadata (gender, district, and linguistic background), raising potential risks for re-identification or misuse. These ethical issues are not acknowledged or mitigated.

Language and cultural sensitivity:
The work frames Hindi speech data as a “testbed for non-Western contexts” but does not engage with potential biases, representational harms, or power asymmetries in using Indian participants to evaluate Western-developed models. There is no reflection on cross-cultural implications or the fairness of comparing spontaneous human speech to machine-generated text.

Dependence on proprietary APIs (e.g., GPT-4.1):
The use of closed-source systems for both generation and evaluation compromises reproducibility and raises transparency concerns. The authors do not address how they ensured consistency, consent, or ethical compliance when using commercial systems to process sensitive linguistic data.

Absence of ethical approval:
Although the study involves human-generated speech data, no mention is made of IRB approval, consent mechanisms, or ethical oversight. This omission is unacceptable for a paper that processes human subject data.

---

### Note · Authors · 2026-01-06

I have read and agree with the venue's withdrawal policy on behalf of myself and my co-authors.